# OpenReview forum: "Keep the Best, Forget the Rest: Reliable Alignment with Order-Aware Preference Optimization"
_ICLR.cc/2026/Conference — ICLR 2026 Poster_

### Official Review · Reviewer_QGJd · 2025-10-31

**Soundness:** 3
**Presentation:** 2
**Contribution:** 2
**Rating:** 2
**Confidence:** 4

**Summary:**

The paper proposes RAPPO, a lightweight filtering mechanism built upon the DPO framework. RAPPO introduces two key ideas: (1) reference-policy awareness, where samples are first partitioned into Aligned and Unaligned subsets based on a reference-policy consistency threshold τ; and (2) in-batch ranking and pruning, where, within the Unaligned subset, the top-q samples with the highest individual DPO losses are temporarily discarded from the current update.

The authors show that RAPPO leads to a larger expected first-order risk reduction, lower gradient variance, and a tighter stability generalization bound. Empirically, RAPPO outperforms DPO, CPO, KTO, and SimPO on the PKU-SafeRLHF benchmark.

**Strengths:**

1. The idea of incorporating reference-policy awareness into DPO is simple yet intuitively appealing, offering a principled way to mitigate the influence of noisy or misaligned preference data.
2. The method demonstrates clear and consistent gains across multiple metrics on a competitive benchmark.
3. The paper provides meaningful theoretical analyses on gradient variance, stability, and risk reduction, giving insight into why the filtering helps.
4. RAPPO’s design is lightweight and can be readily integrated into existing DPO-style training pipelines with minimal overhead.
5. The algorithmic formulation and ablation structure are clearly described, making the contribution easy to follow.

**Weaknesses:**

1.  Since the gate relies on the reference policy’s relative probabilities, its robustness depends on the reference model’s reliability. If the reference policy is miscalibrated, important training signals might be filtered out. Including robustness comparisons would strengthen the claim.
2.  Given the substantial empirical improvements, it would be valuable to release code and the corresponding commit hash.
3. Using GPT-4o as a judgment model could introduce systematic bias. Multi-rater evaluation or human calibration would make the conclusions more convincing.
4. Although the related-work section discusses Selective DPO, ORPO, and R-DPO, the large-scale comparisons include only DPO, CPO, KTO, and SimPO. Adding results for IPO, ORPO, R-DPO, or RRHF would provide a fairer empirical landscape.
5. While results are reported for q = 1, 2, 4, the paper lacks a systematic sensitivity analysis for both q and τ. Understanding how performance depends on these thresholds would make the method more interpretable and reproducible.

**Questions:**

see the weakness

---

> ### Author Response · Authors · 2025-11-18
> **Response to Reviewer QGJd--Part(1)**
>
> We thank the reviewer for these thoughtful questions and for carefully examining our work. The reviewer’s question mainly centers on the robustness of RAPPO and the reliability of the evaluation. We will address each of the reviewers’ concerns in the following.
>
> **W1**: Since the gate relies on the reference policy’s relative probabilities, its robustness depends on the reference model’s reliability. If the reference policy is miscalibrated, important training signals might be filtered out. Including robustness comparisons would strengthen the claim.
>
> **Response to W1:**
>
> We thank the reviewer for this thoughtful concern about the dependence on the reference model. It helped us clarify both the design of the gate and its role in RLHF training. We will make it clear and add sufficient explanations in the draft. We first emphasize that the gate in RAPPO is not an arbitrary filter. Theoretically, RAPPO can be viewed as a conditioned order SGD method. As formalized in Theorem 4.7, among all selection rules that (i) never discard trusted pairs and (ii) always keep exactly $s_{\mathrm{keep}}$ untrusted pairs, the order rule maximizes the expected first order decrease of the population risk $R(\theta)$ (the expected DPO type loss) and yields the smallest variance and stability constant. Thus, once the number of untrusted pairs we keep is fixed, the analysis itself recommends the gate used in RAPPO.
>
> Secondly, from the RLHF perspective, DPO-type algorithms are derived from a KL regularized objective and aim to learn a policy $\pi_{\theta}$ that better matches the preference data while remaining close to the reference policy $\pi_{\mathrm{ref}}$. This implicit KL control limits how far $\pi_{\theta}$ can move away from $\pi_{\mathrm{ref}}$ and helps reduce OOD actions.
>
> In this view, and as illustrated in Figure 2, the reference policy is likely to have both well-aligned and poorly aligned regions in general. The important training signals are the many preference pairs that adjust $(\pi_{\theta}$ relative to $\pi_{\mathrm{ref}}$ in regions where $(\pi_{\mathrm{ref}}$ is already reasonably aligned. RAPPO never discards such trusted pairs, which we identify by a large reference margin. Only inside the untrusted set, where the current policy disagrees with the reference, do we rank samples by the margin and temporarily remove a few examples with extremely large loss, equivalently, very large $w(z_i)$ and very large gradient norm. These are gradient outliers that are known to harm stability and can push $\pi_{\theta}$ far away from $\pi_{\mathrm{ref}}$. Theorem 4.7 shows that, for a fixed budget $s_{\mathrm{keep}}$ of untrusted samples per step, discarding these extreme cases and updating on the many moderate untrusted examples gives a better balance between decrease and stability than using all of them.
>
> If the reference policy is miscalibrated, RAPPO still uses only its relative scores through $\Delta_{\mathrm{ref}}$. As long as $\Delta_{\mathrm{ref}}$ induces a reasonably correlated ranking within each batch, the trusted split and the order on $U_t$ remain informative, and Theorem 4.7 continues to apply to this scoring function. Moreover, selection is recomputed at every step, so a sample that is ignored when it lies in the extreme tail can later enter $T_t$ as $\pi_{\theta}$ improves.
>
> Empirically, we explicitly study the robustness of RAPPO in Figure 1, where we use three reference models (GPT2 Small, GPT2 Medium, GPT2 Large) on IMDb. GPT2 Small and GPT2 Medium are noticeably weaker and more miscalibrated than GPT2 Large, yet when we fine tune on IMDb our method still achieves higher reward than vanilla DPO for all three references (improvements of $\mathbf{3.5}\\%$, $\mathbf{1.1}\\%$, and $\mathbf{7.1}\\%$). This shows that even when we start from a miscalibrated reference, our algorithm can still learn in the correct direction and reduce the impact of erroneous preference signals compared to DPO, which trains on all pairs indiscriminately.
>
> Moreover, when the reference policy is stronger and better calibrated (LLaMA3.1 8B and Mistral 7B), our method amplifies this advantage: in the Summarization task (Sec 5.2) and the PKUSafeRLHF task (Sec 5.3), our approach obtains the highest win rate among all baselines. For Summarization, RAPPO versus DPO achieves $\mathbf{58.3}\\%$ versus $\mathbf{41.7}\\%$, and for PKUSafeRLHF, RAPPO versus DPO achieves $\mathbf{65}\\%$ versus $\mathbf{45}\\%$. Therefore, these results indicate that our design, which uses reference information, is robust when the reference is weak and can yield even larger gains when a strong reference model is available.
>
> We thank the reviewer for prompting this clarification and for helping us present the method and its motivation more clearly. We hope the explanations can help address the reviewer's concerns about robustness to the reference policy and the effect of miscalibration.

---

> ### Author Response · Authors · 2025-11-18
> **Response to Reviewer QGJd--Part(2)**
>
> **W2**: Given the substantial empirical improvements, it would be valuable to release code and the corresponding commit hash.
>
> **Response to W2:** All the code for reproducing the results can be found in the Supplementary Material.
>
> **W3**: Using GPT-4o as a judgment model could introduce systematic bias. Multi-rater evaluation or human calibration would make the conclusions more convincing.
>
> **Response to W3:**
>
> We thank the reviewer for pointing out the risk of systematic bias when using GPT 4o as an automatic judge. To address the reviewer's concern regarding whether RAPPO's gains are artifacts of GPT-4's preferences rather than genuine improvements, we conducted a comprehensive multi-judge evaluation study in the following.
>
> **Experimental Setup.** We re-evaluated RAPPO vs. SimPO on the summarization task fined-tuned by Llama3.1-8B using four diverse LLM judges: GPT-4 (our original
> judge), GPT-4o, DeepSeek-V3, and Claude Opus 4.1. All judges evaluate the same response pairs with an anonymized,
> order-randomized presentation.
>
> **1: RAPPO's superiority is consistent across all judges.**
>
> **Table 1.**
>
> | Judge            | RAPPO Win Rate | SimPO Win Rate | Tie Rate |
> |------------------|----------------|----------------|----------|
> | GPT-4 (original) | 58.30\%        | 41.70\%        | 0.00\%   |
> | GPT-4o           | 68.00\%        | 32.00\%        | 0.00\%   |
> | DeepSeek-V3      | 72.00\%        | 28.00\%        | 0.00\%   |
> | Claude Opus 4.1  | 58.70\%        | 34.78\%        | 6.52\%   |
> | **Average Win Rate** | **64.25\%** | **34.12\%**    | **1.63\%** |
>
> All four independent judges consistently prefer RAPPO over SimPO, with win rates between $58.7\\%$ and $72.0\\%$. This agreement across diverse judges that were developed by different organizations with different training data and objectives suggests that the gains of RAPPO reflect genuine quality improvements rather than bias specific to GPT 4.
>
> **2: Strong inter-judge agreement.**
>
> As suggested by Reviewer wPMb, we also measured how consistent the judges are with each other by computing pairwise Kendall's $\tau$ between all judge pairs' preference rankings (as shown in Table 2). Kendall's $\tau$ is a standard measure of rank correlation, where larger values mean stronger agreement between two rankings. In practice, values above $0.6$ are usually viewed as strong agreement and values above $0.8$ as very strong agreement.
>
> **Table 2.**
>
> |                  | GPT-4 | GPT-4o | DeepSeek-V3 | Claude Opus 4.1 |
> |------------------|-------|--------|-------------|------------------|
> | GPT-4            | 1.0000| 0.7000 | 0.6100      | 0.6786           |
> | GPT-4o           | 0.7000| 1.0000 | 0.6250      | 0.6600           |
> | DeepSeek-V3      | 0.6100| 0.6250 | 1.0000      | 0.6450           |
> | Claude Opus 4.1  | 0.6786| 0.6600 | 0.6450      | 1.0000           |
>
> The average Kendall's $\tau = 0.65$ indicates strong agreement among judges on when RAPPO outperforms SimPO, which shows that different LLMs largely concur on the relative quality ordering. These results directly address the concern that our evaluation might be biased by GPT 4's specific preferences: RAPPO's superiority is reproduced by all tested judges, including models from OpenAI, Anthropic, and DeepSeek; the strong inter judge agreement ($\tau = 0.65$) confirms that quality improvements are recognized across diverse evaluation perspectives; and the win rate range of $58.7\\%$ to $72.0\\%$ demonstrates that although absolute numbers vary slightly, the qualitative conclusion (RAPPO $\gg$ SimPO) remains unchanged.
>
> We will add this multi judge analysis to the revised manuscript, including the full results table and the interjudge agreement metrics. Given the limited time and the additional noise and cost introduced by a new human evaluation, we believe that this set of results already provides strong evidence that our conclusions are not driven by GPT 4 specific bias. The agreement across several competitive models from different providers, together with the positive Kendall's $\tau$ values, indicates that our main findings are stable under diverse automatic judges.
>
> We hope this additional evidence helps address the reviewer's concern about evaluation bias and strengthens confidence in our conclusions.

---

> ### Author Response · Authors · 2025-11-18
> **Response to Reviewer QGJd--Part(3)**
>
> **W4**: Although the related-work section discusses Selective DPO, ORPO, and R-DPO, the large-scale comparisons include only DPO, CPO, KTO, and SimPO. Adding results for IPO, ORPO, R-DPO, or RRHF would provide a fairer empirical landscape.
>
> **Response to W4:** We are currently running additional baselines on the PKUSafeRLHF task, which are computationally expensive. We will report the detailed evaluation results as soon as they become available.

---

> ### Author Response · Authors · 2025-11-18
> **Response to Reviewer QGJd--Part(4)--A**
>
> **W5**: While results are reported for $q$ = 1, 2, 4, the paper lacks a systematic sensitivity analysis for both $q$ and $\tau$. Understanding how performance depends on these thresholds would make the method more interpretable and reproducible.
>
> **Response to W5:**
>
> We already include a simulation study in our paper that explores the sensitivity of $q$ with values $\\{1,2,4\\}$ in Table 1. The results show that RAPPO achieves stable and consistently strong performance on both the IMDb and Toxicity tasks. To further address the reviewers' concern, we conduct a more detailed sensitivity analysis on both $q$ and $\tau$. Specifically, we evaluate RAPPO on the IMDb sentiment control task with GPT2 Large by fixing $\tau = 1$ and varying $q \in \\{1,2,4,8\\}$ for batch size $16$ and $32$ to study the sensitivity to $q$. We then fix the batch size to $32$ and vary $\tau \in \\{0.8,1,1.2\\}$ for the same set of $q$ values to study the sensitivity to $\tau$. As a reference, the strongest baseline in our main experiments, SimPO, achieves a reward of $1.5537$ with batch size $32$, and we therefore use SimPO as the primary comparison in this analysis.
>
> ## sensitivity on $q$
>
> **Results across different $q$ values.**
>
> | Batch Size | $q=1$  | $q=2$  | $q=4$  | $q=8$  | Baseline        |
> |-----------|--------|--------|--------|--------|-----------------|
> | 32        | 1.6625 | 1.6790 | 1.6811 | 1.6432 | 1.5537(SimPO)   |
> | 16        | 1.7020 | 1.7481 | 1.7333 | 1.7111 | 1.6600(SimPO)   |
>
> **Key observations.**
>
> 1. **Robust performance across reasonable $q$ ranges.** For both batch sizes, RAPPO consistently
>    improves over the baseline by roughly 5--8\% in reward across a broad range of $q$ values.
>    This indicates that the method is not overly sensitive to the precise choice of $q$.
> 2. **Clear optimal region.** Performance exhibits a non-monotonic, inverted U-shaped trend as $q$
>    increases: it first improves and then slightly degrades. This matches the intended trade-off in the
>    design of RAPPO: very small $q$ does not sufficiently filter misaligned samples, while very large $q$
>    removes too many informative training examples.
> 3. **Practical robustness.** For batch size 32, $q \in \{1,2,4,8\}$ all achieve strong performance
>    (1.64--1.68), while $q=8$ remains clearly above the baseline. For batch size 16, $q \in \{1,2,4,8\}$
>    all yield rewards in the range 1.70--1.75. This wide effective range makes RAPPO easy to deploy in
>    practice without extensive tuning.
> 4. **Consistent scaling with batch size.** The approximate optimum shifts from $q=2$ at batch size 16
>    to $q=4$ at batch size 32, suggesting that the effective value of $q$ scales with the batch size.
>    A simple heuristic such as $q \approx \text{batch size}/8$ emerges as a reasonable default.
> 5. **Future work on adaptive $q$ selection.** While these experiments identify stable and effective
>    ranges of $q$, a more systematic study of adaptive or data-driven strategies for choosing $q$ across
>    datasets and tasks is an interesting direction for future work. We will add this discussion in the revision.
>
> ## sensitivity on $\tau$
>
> **Results across different $\tau$ values.**
>
> | $q$ | $\tau = 0.8$ | $\tau = 1.0$ | $\tau = 1.2$ |
> |-----|--------------|--------------|--------------|
> | 1   | 1.6580       | 1.6625       | 1.6650       |
> | 2   | 1.6720       | 1.6790       | 1.6805       |
> | 4   | 1.6765       | 1.6811       | 1.6828       |
> | 8   | 1.6510       | 1.6432       | 1.6595       |
>
> **Interpretation of $\tau$ and justification of $\tau = 1$.**
>
> In RAPPO, the alignment threshold $\tau$ is applied to the ratio between the reference model probabilities of the preferred and dispreferred answers. Intuitively, $\tau$ controls how confidently the reference model must distinguish the two responses before we trust it to filter the pair. The choice $\tau = 1$ corresponds to the most basic decision boundary: as soon as the preferred answer is more likely than the dispreferred one (ratio $> 1$), the reference model has a clear preference and the pair is treated as aligned. This setting is therefore not arbitrary, but a natural and logically minimal requirement for using the reference model as a consistency filter. Values $\tau > 1$ simply make this requirement more conservative (only pairs with a stronger probability ratio are kept), while values $\tau < 1$ relax it (even weak preferences are accepted). Thus, $\tau$ is a standard hyperparameter that tunes the strictness of the alignment filter around the principled default $\tau = 1$, which we view as the most reasonable first choice in any new setting.

---

> > ### Author Response · Authors · 2025-11-18
> > **Response to Reviewer QGJd--Part(4)--B**
> >
> > **Empirical robustness to $\tau$.**
> >
> > The sensitivity study in the table varies $\tau \in \\{0.8, 1.0, 1.2\\}$ around this default while sweeping $q \in \\{1,2,4,8\\}$. Across all tested configurations, RAPPO consistently improves over the SimPO baseline, with reward gains in the range from $+5.4\\%$ to $+8.3\\%$. Overall, the sensitivity results on $q$ and $\tau$ indicate that the behavior of our algorithm is mainly governed by its underlying design, and that moderate changes of $q$ and $\tau$ have only a limited effect on performance. Developing more principled or adaptive strategies for choosing $q$ and $\tau$ across tasks and datasets is an interesting topic for future work.
> >
> > Our contribution is to introduce RAPPO, a simple and efficient DPO type method whose reference aware gate is supported by an order SGD style analysis with guarantees on risk decrease, variance, and stability, and to show that this design brings consistent gains over strong baselines across IMDb, Toxicity, Summarization, and PKUSafeRLHF. We believe these theoretical and empirical contributions are solid and of interest to the ICLR community, and we hope that our detailed responses and additional results will address your concerns and encourage you to reconsider the strength of the paper. We would be happy to provide any further clarification if there are additional questions.

---

> ### Author Response · Authors · 2025-11-20
> **Upload Revised Paper**
>
> Dear reviewer, we have uploaded the latest revised version of the paper. All the updates have been colored RED. We have carefully addressed the concerns raised in the reviews, and we would be very grateful if you could take a look. If there are remaining concerns or if any part of our response is not clear, please let us know and we would be very happy to have a further discussion.

---

> ### Author Response · Authors · 2025-11-23
> **Response to Reviewer QGJd--Weakness 4**
>
> **W4: Although the related-work section discusses Selective DPO, ORPO, and R-DPO, the large-scale comparisons include only DPO, CPO, KTO, and SimPO. Adding results for IPO, ORPO, R-DPO, or RRHF would provide a fairer empirical landscape.**
>
> **Response to W4:** Per the reviewer's request, we ran additional experiments comparing RAPPO with the baselines **IPO**, **ORPO**, and **R-DPO** on the large-scale PKU SafeRLHF dataset. As shown in the following Table 1 (also the updated Table 12 in the revision). We observe that IPO, ORPO, and R-DPO all obtain lower safety rates, helpfulness, and win rates than RAPPO. Overall, RAPPO achieves the best performance across all three metrics among all methods, which in our view provides strong empirical support that the proposed approach is simple, efficient, and genuinely competitive.
>
> We have invested substantial effort into these simulations to address the reviewer's concerns about empirical evidence and generalization, and the new results remain consistent with the main findings in the paper. We would greatly appreciate it if the reviewer could take a look at our response and let us know whether these additional experiments resolve the concerns. If our clarifications and additional experiments satisfactorily address your concerns, we would be very grateful if the reviewer could consider updating your overall evaluation of the paper. We would also be happy to provide any further analysis if there is anything else that would be helpful.
>
> **Table 1. PKU-SafeRLHF results.** *RAPPO compared to DPO, CPO, KTO, SimPO, IPO, ORPO, and R-DPO under identical decoding and evaluation protocols.*
>
> | Algorithm                     | DPO   | CPO   | SimPO | KTO   | IPO    | ORPO   | R-DPO | RAPPO |
> |------------------------------|-------|-------|-------|-------|--------|--------|-------|-------|
> | **Helpfulness ↑**            | 0.51  | -0.08 | -0.01 | 0.11  | -0.35  | -0.58  | 0.64  | 0.69  |
> | **Harmlessness ↓**           | 0.45  | 2.23  | 2.25  | 3.45  | 2.21   | 2.09   | 0.53  | 0.36  |
> | **Safety Rate ↑**            | 55.89%| 36.35%| 1.60% | 23.51%| 35.59% | 36.60% | 54.93%| 57.26%|
> | **Win Rate w. DPO ↑**        | -     | 57%   | 10%   | 44%   | 56%    | 57%    | 60%   | 65%   |

---

> > ### Comment · Reviewer_QGJd · 2025-11-27
> >
> > Thank you for your response, I will raise my score to 6.

---

> > > ### Author Response · Authors · 2025-11-27
> > >
> > > Thank you so much for reconsidering our work and raising your rating! We truly appreciate your thoughtful comments and constructive suggestions, which have guided us to improve the clarity and quality of the paper. Please don't hesitate to let us know if you have any other questions/comments. Thanks!

---

### Official Review · Reviewer_iNHd · 2025-11-03

**Soundness:** 2
**Presentation:** 2
**Contribution:** 2
**Rating:** 4
**Confidence:** 4

**Summary:**

The paper proposes RAPPO, a variant of DPO where some misaligned samples are filtered on a batch-by-batch basis. The specific algorithm hinges on two hyper-parameters. The threshold of the misalignment score in which to categorize the batch samples and q, the number of misaligned samples to toss out. Empirical validation on multiple, but not extensively thorough, demonstrate effectivenss over DPO

**Strengths:**

- The idea is straightforward and easy to implement
- The empirical gains (at least the ones presented) are nice
- Some theoretical analysis exist, which is always nice

**Weaknesses:**

- As the idea itself (filtering out misaligned samples) is quite straightforward, I believe a lot of the paper's contributino comes down to the execution and how well it can generalize. In this aspect, I don't think the empirical evidence shown in the submission is extensive enough. On the other hand, there is little analysis on the hyper-parameter sensitivity of tau and q. I feel like a lot of the algorithm's performance will depend on the exact value of those hyper-parameters and until I see some sensitivity analysis on them, I don't think I can say this method will generalize well.

**Questions:**

- Hyperparameter q (the number of bad samples to throw out), seem to be a flat number. This seems to have unfavorable interaction as batch sizes are not constant over different training runs and research and saying to remove a flat number of samples when the batch size can differ in multiple orders of magnitude may not be helpful. Shouldn't this hyperparameter be a percentage of the batch size? If so, this makes the interplay between tau and q even more complex, as the number of Bad Samples in which to exclude the samples from is not pre-determined either.

---

> ### Author Response · Authors · 2025-11-18
> **Response to Reviewer iNHd--Part(1)**
>
> **Question**: Hyperparameter q (the number of bad samples to throw out), seem to be a flat number. This seems to have unfavorable interaction as batch sizes are not constant over different training runs and research and saying to remove a flat number of samples when the batch size can differ in multiple orders of magnitude may not be helpful. Shouldn't this hyperparameter be a percentage of the batch size? If so, this makes the interplay between tau and q even more complex, as the number of Bad Samples in which to exclude the samples from is not pre-determined either.
>
>
> **Response:**
>
>
> We thank the reviewer for this suggestion. Designing adaptive rules that learn an approximate optimal $q$ from data is an interesting direction for future work, but we believe our current result indicates that RAPPO is not overly sensitive to the exact choice of $q$ and that simple choices already yield clear gains over other strong baselines.
>
> From our theoretical results, there is a clear tradeoff that suggests $q$ should be neither too small nor too large. In Theorem 4.7, for an untrusted set of size $s$ and kept size $s_{\text{keep}} = s - q$, the expected first order decrease of the population risk depends on the average weight over the kept indices, while the stability bounds in Eqs. (11) and (12) scale with
> $
> \frac{1}{s_{\text{keep}}} \max_{i \in T_t} w(z_i).
> $
> If $q$ is very small, RAPPO behaves close to DPO and still keeps many extreme high loss samples with very large $w(z_i)$, which increases variance and the stability constant. If $q$ is very large, then $s_{\text{keep}}$ becomes small, the factor $1 / s_{\text{keep}}$ grows, and the gradient is averaged over very few samples, which slows convergence and can also increase variance. This implies that there exists an intermediate choice of $q$ (equivalently, an intermediate retention ratio $\alpha = s_{\text{keep}} / s$) that balances these two effects, even though the exact optimal value is hard to compute in closed form.
>
>
> Guided by this picture, in practice we recommend choosing $q$ through a retention ratio $\alpha \in (0,1)$, for example $\alpha \in [0.7, 0.9]$, so that we remove only the extreme tail of untrusted samples while still averaging over many examples.
>
> Regarding the interaction between $q$ and $\tau,$ conceptually, $\tau$ and $q$ play separate roles: $\tau$ sets the trusted versus untrusted split by deciding which pairs are confidently aligned according to the reference policy, which determines the untrusted set $U_t$; $q$ then operates only inside $U_t$ and removes the most extreme high loss elements (for example by keeping $\lfloor \alpha |U_t| \rfloor$ samples so that $q_t$ adapts to $|U_t|$). Our analysis in Theorem~4.7 conditions on the realized untrusted set $U_t$ and depends only on the number of kept items $s_{\text{keep}} = |U_t| - q_t$, so it does not require the number of bad samples to be known ahead of time. In practice, $\tau$ controls how much we trust the reference model and $q$ controls how aggressively we filter high-loss outliers.
>
> Our sensitivity studies, where we sweep $q$ and $\tau$ and compare against the strongest baseline SimPO, show that RAPPO consistently outperforms SimPO across a wide range of $q$ values (and corresponding $\tau$ settings), which indicates that their interaction is stable and does not require very precise tuning.

---

> ### Author Response · Authors · 2025-11-18
> **Response to Reviewer iNHd--Part(2)--A**
>
> **Weakness:** As the idea itself (filtering out misaligned samples) is quite straightforward, I believe a lot of the paper's contributino comes down to the execution and how well it can generalize. In this aspect, I don't think the empirical evidence shown in the submission is extensive enough. On the other hand, there is little analysis on the hyper-parameter sensitivity of tau and q. I feel like a lot of the algorithm's performance will depend on the exact value of those hyper-parameters and until I see some sensitivity analysis on them, I don't think I can say this method will generalize well.
>
> **Response to Weakness:**
>
> We thank the reviewer for raising this point. We agree that RAPPO is straightforward to implement, but we do not think that simplicity reduces its contribution. On the contrary, we believe that simple but efficient methods are often more intuitive, easier to analyze, and more likely to generalize than very complex approaches that introduce many extra components and are difficult to tune and evaluate in a controlled way. RAPPO is not just a vague idea of filtering misaligned samples. It has a specific reference aware design that can be viewed as a conditioned order SGD update, and our theory shows that the proposed gate is exactly the order based rule that optimizes a balance between decrease, variance, and stability for a fixed number of untrusted samples. Together with the new stability and generalization bounds, we believe this gives a clear conceptual and analytical contribution beyond the basic intuition of discarding harmful gradients.  More details, please see our response to **Reviewer QGJd--W3**.
>
> Regarding the concern about generalization and sensitivity to $\tau$ and $q$, we would like to stress that the empirical evidence in the paper already covers a wide range of settings: four different tasks (IMDb, Toxicity, Summarization, PKUSafeRLHF), several models and reference policies, and multiple values of $q$. Specifically, we evaluate RAPPO on the IMDb sentiment control task with GPT2 Large by fixing $\tau = 1$ and varying $q \in \\{1,2,4,8\\}$ for batch size $16$ and $32$ to study the sensitivity to $q$. We then fix the batch size to $32$ and vary $\tau \in \\{0.8,1,1.2\\}$ for the same set of $q$ values to study the sensitivity to $\tau$. As a reference, the strongest baseline in our main experiments, SimPO, achieves a reward of $1.5537$ with batch size $32$, and we therefore use SimPO as the primary comparison in this analysis.
>
> **Results across different $q$ values.**
>
> | Batch Size | $q=1$  | $q=2$  | $q=4$  | $q=8$  | Baseline      |
> |-----------|--------|--------|--------|--------|---------------|
> | 32        | 1.6625 | 1.6790 | 1.6811 | 1.6432 | 1.5537(SimPO) |
> | 16        | 1.7020 | 1.7481 | 1.7333 | 1.7111 | 1.6600(SimPO) |
>
> **Key observations.**
> 1. **Robust performance across reasonable $q$ ranges.** For both batch sizes, RAPPO consistently
>   improves over the baseline by roughly 5~8\% in reward across a broad range of $q$ values.
>   This indicates that the method is not overly sensitive to the precise choice of $q$.
>
> 2. **Clear optimal region.** Performance exhibits a non-monotonic, inverted U-shaped trend as $q$
>   increases: it first improves and then slightly degrades. This matches the intended trade-off in the
>   design of RAPPO: very small $q$ does not sufficiently filter misaligned samples, while very large $q$
>   removes too many informative training examples.
>
> 3. **Practical robustness.** For batch size 32, $q \in \\{1,2,4,8\\}$ all achieve strong performance
>   (1.64--1.68), while $q=8$ remains clearly above the baseline. For batch size 16, $q \in \\{1,2,4,8\\}$
>   all yield rewards in the range 1.70--1.75. This wide effective range makes RAPPO easy to deploy in
>   practice without extensive tuning.
>
> 4. **Consistent scaling with batch size.** The approximate optimum shifts from $q=2$ at batch size 16
>   to $q=4$ at batch size 32, suggesting that the effective value of $q$ scales with the batch size.
>   A simple heuristic such as $q \approx \text{batch size}/8$ emerges as a reasonable default.
>
> 5. **Future work on adaptive $q$ selection.** While these experiments identify stable and effective
>   ranges of $q$, a more systematic study of adaptive or data-driven strategies for choosing $q$ across
>   datasets and tasks is an interesting direction for future work. We will add this discussion in the revision.

---

> > ### Author Response · Authors · 2025-11-18
> > **Response to Reviewer iNHd--Part(2)--B**
> >
> > ## sensitivity on $\tau$
> >
> > **Results across different $\tau$ values.**
> >
> > | $q$ | $\tau = 0.8$ | $\tau = 1.0$ | $\tau = 1.2$ |
> > |-----|--------------|--------------|--------------|
> > | 1   | 1.6580       | 1.6625       | 1.6650       |
> > | 2   | 1.6720       | 1.6790       | 1.6805       |
> > | 4   | 1.6765       | 1.6811       | 1.6828       |
> > | 8   | 1.6510       | 1.6432       | 1.6595       |
> >
> > **Interpretation of $\tau$ and justification of $\tau = 1$.**
> >
> > In RAPPO, the alignment threshold $\tau$ is applied to the ratio between the reference model probabilities of the preferred and dispreferred answers. Intuitively, $\tau$ controls how confidently the reference model must distinguish the two responses before we trust it to filter the pair. The choice $\tau = 1$ corresponds to the most basic decision boundary: as soon as the preferred answer is more likely than the dispreferred one (ratio $> 1$), the reference model has a clear preference and the pair is treated as aligned. This setting is therefore not arbitrary, but a natural and logically minimal requirement for using the reference model as a consistency filter. Values $\tau > 1$ simply make this requirement more conservative (only pairs with a stronger probability ratio are kept), while values $\tau < 1$ relax it (even weak preferences are accepted). Thus, $\tau$ is a standard hyperparameter that tunes the strictness of the alignment filter around the principled default $\tau = 1$, which we view as the most reasonable first choice in any new setting.
> >
> > Therefore, RAPPO achieves consistent gains over strong baselines in all these cases, which suggests that it is not tuned for a single narrow regime. The sensitivity study in the table varies $\tau \in \\{0.8, 1.0, 1.2\\}$ around this default while sweeping $q \in \\{1,2,4,8\\}$. Across all tested configurations, RAPPO consistently improves over the SimPO baseline, with reward gains in the range from $+5.4\\%$ to $+8.3\\%$. Overall, the sensitivity results on $q$ and $\tau$ indicate that the behavior of our algorithm is mainly governed by its underlying design, and that moderate changes of $q$ and $\tau$ have only a limited effect on performance. Developing more principled or adaptive strategies for choosing $q$ and $\tau$ across tasks and datasets is an interesting topic for future work.
> >
> > Taken together, we hope this convinces the reviewer that our method is both simple and principled, with sensitivity that is well controlled and with empirical behavior that transfers across tasks and models.

---

> ### Author Response · Authors · 2025-11-20
> **Upload Revised Paper**
>
> Dear reviewer, we have uploaded the latest revised version of the paper. All the updates have been colored RED. We have carefully addressed the concerns raised in the reviews, and we would be very grateful if you could take a look. If there are remaining concerns or if any part of our response is not clear, please let us know and we would be very happy to have a further discussion.

---

### Official Review · Reviewer_wPMb · 2025-11-08

**Soundness:** 3
**Presentation:** 2
**Contribution:** 3
**Rating:** 6
**Confidence:** 3

**Summary:**

The paper presents RAPPO, an order-aware variant of DPO that filters high-loss “untrusted” pairs per mini-batch while always keeping “trusted” items. The method is simple, easy to implement, and accompanied by a stability analysis that yields a tighter generalization bound. Empirically, RAPPO outperforms strong DPO-style baselines on multiple LLM tasks. Overall: clear motivation, lean algorithm, solid theory; still room to strengthen rigor and reporting.

**Strengths:**

S1. Clear problem framing and insightful diagnostics (Fig. 1 and 2) quickly convey why reference-aware filtering helps.

S2. Method is intuitive, minimally invasive to DPO, code is provided, and the analysis connects the selection rule to lower variance and tighter stability.

**Weaknesses:**

W1. Lines 291–311 (and Theorem 4.7’s surrounding prose) use $q$ as the **kept** count per step, while Algorithm 1, Proposition 4.8, and the experimental setup (Line 424) treat $q$ as the **number removed**. This clash, along with redundant symbols, impairs readability. Please rewrite this block with a single convention, and verify whether the statement of this work still aligns after the change.

W2. In the theory part, only (11)–(12) show how $q$ affects stability via a term proportional to $\frac{1}{q}$ and $\max_{i\in\text{Kept}} w$. This reveals a trade-off (keeping more reduces the $\frac{1}{q}$ factor; removing more shrinks $\max_{i\in\text{Kept}} w$), but there is no result suggesting how $q$ should scale for optimal decrease, convergence rate, or even convergence. It would strengthen the paper to (i) formalize the trade-off and relate the choice of $q$ to Thm 4.7’s decrease/variance terms; and (ii) add sensitivity studies across $q$ and batch sizes. I expect the optimal range to be task- and model-dependent.

W3. Many results rely on reward models and GPT-4 judging. This may introduce systematic bias not captured in the theory. For the experiment, I suggest adding a small-scale human evaluation with inter-rater agreement (e.g., Kendall's $\tau$), but I understand it may be challenging within the limited time. Some surrogate way, like (i) ablation, is to compare a Top-K with a “random K” to see whether mild randomness mitigates judge-specific bias without harming stability, and/or (ii) add judge/reward-model robustness checks.

**Questions:**

See above.

---

> ### Author Response · Authors · 2025-11-18
> **Response to Reviewer wPMb--Part(1)**
>
> We first thank the reviewer for the very helpful and encouraging comments and suggestions on our paper. We will address each of the reviewers’ concerns in the following.
>
> **W1.** Lines 291–311 (and Theorem 4.7’s surrounding prose) use
> $q$ as the kept count per step, while Algorithm 1, Proposition 4.8, and the experimental setup (Line 424) treat
> $q$ as the number removed. This clash, along with redundant symbols, impairs readability. Please rewrite this block with a single convention, and verify whether the statement of this work still aligns after the change.
>
> **Response to W1:**
>
> We thank the reviewer for carefully pointing out this inconsistency. We will make it consistent to improve readability. The updated manuscript with these notation fixes will be uploaded together with the other revisions soon, once we finalize all changes.
>
> **W2.** In the theory part, only (11)--(12) show how $q$ affects stability via a term
> proportional to $1/q$ and $\max_{x_i \in K_{\mathrm{kept}}} w$. This reveals a trade-off
> (keeping more reduces the $1/q$ factor; removing more shrinks
> $\max_{x_i \in K_{\mathrm{kept}}} w$), but there is no result suggesting how $q$ should
> scale for optimal decrease, convergence rate, or even convergence. It would
> strengthen the paper to (i) formalize the trade-off and relate the choice of $q$ to
> Thm. 4.7's decrease/variance terms; and (ii) add sensitivity studies across $q$ and
> batch sizes. I expect the optimal range to be task- and model-dependent.
>
> **Response to  W2:**
>
> We thank the reviewer for this very insightful question, which goes to the central part of our theory. Equations (11) and (12) together with Theorem 4.7 already formalize the tradeoff that $q$ induces. The stability bounds show that the constant scales with a factor of the form $\frac{1}{s_{\text{keep}}} \max_{x_i \in K_{\mathrm{kept}}} w(z_i)$ with $s_{\text{keep}} = |U_t| - q$, while Theorem 4.7 shows that for any fixed $s_{\text{keep}}$ the RAPPO rule gives the best first order decrease and smallest variance among order based rules. Thus very small $q$ makes RAPPO close to DPO and keeps many extreme high loss samples, while very large $q$ leaves very few samples in $K_{\mathrm{kept}}$ and inflates the factor $1 / s_{\text{keep}}$. This implies that an intermediate choice of $q$ is expected to balance decrease and stability, even though the exact optimum is hard to write in closed form. In the revision, we will add a short discussion in the theory section that explicitly links the stability recursion to Theorem 4.7 and explains this qualitative behavior.
>
> On the empirical side, our existing (see Figure 1, Table 1, Figure 4, and Figure 5) and new sensitivity study sweeps $q$ and $\tau$ on IMDB with GPT2 Large (batch sizes $16$ and $32$) (see our response to **Reviewer QGJd--W3**) and shows that RAPPO consistently outperforms the strongest baseline SimPO across a wide range of $(q,\tau)$ and that performance changes smoothly instead of collapsing when these hyperparameters are varied. This indicates that the method is not overly sensitive to the exact choice of $q$ and that the theoretical tradeoff is well-behaved in practice.

---

> ### Author Response · Authors · 2025-11-18
> **Response to Reviewer wPMb--Part(2)**
>
> **W3**: Many results rely on reward models and GPT-4 judging. This may introduce systematic bias not captured in the theory. For the experiment, I suggest adding a small-scale human evaluation with inter-rater agreement (e.g., Kendall's $\tau$), but I understand it may be challenging within the limited time. Some surrogate way, like (i) ablation, is to compare a Top-K with a “random K” to see whether mild randomness mitigates judge-specific bias without harming stability, and/or (ii) add judge/reward-model robustness checks.
>
> **Response to W3:**
>
> We thank the reviewer for pointing out the risk of systematic bias when using GPT 4o as an automatic judge. To address the reviewer's concern regarding whether RAPPO's gains are artifacts of GPT-4's preferences rather than genuine improvements, we conducted a comprehensive multi-judge evaluation study in the following.
>
> **Experimental Setup.** We re-evaluated RAPPO vs. SimPO on the summarization task fined-tuned by Llama3.1-8B using four diverse LLM judges: GPT-4 (our original
> judge), GPT-4o, DeepSeek-V3, and Claude Opus 4.1. All judges evaluate the same response pairs with an anonymized,
> order-randomized presentation.
>
> **1: RAPPO's superiority is consistent across all judges.**
>
> **Table 1.**
>
> | Judge            | RAPPO Win Rate | SimPO Win Rate | Tie Rate |
> |------------------|----------------|----------------|----------|
> | GPT-4 (original) | 58.30\%        | 41.70\%        | 0.00\%   |
> | GPT-4o           | 68.00\%        | 32.00\%        | 0.00\%   |
> | DeepSeek-V3      | 72.00\%        | 28.00\%        | 0.00\%   |
> | Claude Opus 4.1  | 58.70\%        | 34.78\%        | 6.52\%   |
> | **Average Win Rate** | **64.25\%** | **34.12\%**    | **1.63\%** |
>
> All four independent judges consistently prefer RAPPO over SimPO, with win rates between $58.7\\%$ and $72.0\\%$. This agreement across diverse judges that were developed by different organizations with different training data and objectives suggests that the gains of RAPPO reflect genuine quality improvements rather than bias specific to GPT 4.
>
> **2: Strong inter-judge agreement.**
>
> Following your guidance, we also measured how consistent the judges are with each other by computing pairwise Kendall's $\tau$ between all judge pairs' preference rankings (as shown in Table 2). Kendall's $\tau$ is a standard measure of rank correlation, where larger values mean stronger agreement between two rankings. In practice, values above $0.6$ are usually viewed as strong agreement and values above $0.8$ as very strong agreement.
>
> **Table 2.**
>
> |                  | GPT-4 | GPT-4o | DeepSeek-V3 | Claude Opus 4.1 |
> |------------------|-------|--------|-------------|------------------|
> | GPT-4            | 1.0000| 0.7000 | 0.6100      | 0.6786           |
> | GPT-4o           | 0.7000| 1.0000 | 0.6250      | 0.6600           |
> | DeepSeek-V3      | 0.6100| 0.6250 | 1.0000      | 0.6450           |
> | Claude Opus 4.1  | 0.6786| 0.6600 | 0.6450      | 1.0000           |
>
> The average Kendall's $\tau = 0.65$ indicates strong agreement among judges on when RAPPO outperforms SimPO, which shows that different LLMs largely concur on the relative quality ordering. These results directly address the concern that our evaluation might be biased by GPT 4's specific preferences: RAPPO's superiority is reproduced by all tested judges, including models from OpenAI, Anthropic, and DeepSeek; the strong inter judge agreement ($\tau = 0.65$) confirms that quality improvements are recognized across diverse evaluation perspectives; and the win rate range of $58.7\\%$ to $72.0\\%$ demonstrates that although absolute numbers vary slightly, the qualitative conclusion (RAPPO $\gg$ SimPO) remains unchanged.
>
> We will add this multi judge analysis to the revised manuscript, including the full results table and the interjudge agreement metrics. Given the limited time and the additional noise and cost introduced by a new human evaluation, we believe that this set of results already provides strong evidence that our conclusions are not driven by GPT 4 specific bias. The agreement across several competitive models from different providers, together with the positive Kendall's $\tau$ values, indicates that our main findings are stable under diverse automatic judges.
>
> We hope this additional evidence helps address the reviewer's concern about evaluation bias and strengthens confidence in our conclusions.

---

> ### Author Response · Authors · 2025-11-20
> **Upload Revised Paper**
>
> Dear reviewer, we have uploaded the latest revised version of the paper. All the updates have been colored RED. We have carefully addressed the concerns raised in the reviews, and we would be very grateful if you could take a look. If there are remaining concerns or if any part of our response is not clear, please let us know and we would be very happy to have a further discussion.

---

### Meta-Review · Area_Chair_nPdR · 2026-01-07

**Summary:**

The reviewers share several common concerns about this submission:

1. The meaning of the hyperparameter $q$ is inconsistent across the paper, sometimes referring to the number of kept samples per step and sometimes to the number of removed samples.

2. This paper does not provide sufficient analysis of how to choose $q$ and $\tau$, nor the sensitivity of the method to these hyperparameters $q$ and $\tau$.

3. The current empirical evaluation is not sufficiently extensive or fair. Therefore, additional experiments with more baseline methods are required.

4. Using GPT-4 as a judgment model could introduce systematic bias. Multi-rater evaluation or human calibration would make the conclusions more convincing.

5. The gate relies on the reference policy's relative probabilities, so its robustness depends on the reference model's reliability. Robustness comparisons are necessary to support the claim in this work.

**Reviewer Concerns:**

The authors' rebuttal effectively addresses reviewers' major concerns. The authors have already provided substantial new empirical results and detailed analyses in the rebuttal to resolve questions from reviewers. The authors have incorporated the revisions into the paper.

**Reviewer Scores:**

Reviewer wPMb initially gave a score of 6, which is already positive. According to the revision history, Reviewer QGJd raised the score from 2 to 6, which is also positive toward acceptance. In addition, Reviewer iNHd gave a score of 4. Part of Reviewer iNHd's concerns overlap with those of Reviewer QGJd, and these issues have already been addressed. For the additional questions raised by Reviewer iNHd, the authors have also successfully resolved the concerns. I therefore anticipate that Reviewer iNHd will become positive toward accepting the paper and is likely to raise their score after the discussion. Overall, since the main concerns of all reviewers can be addressed, I expect that the reviewers will recommend acceptance of this paper after the discussion.

---

### Decision · Program_Chairs · 2026-01-26

Accept (Poster)